# Anti-phase boundary accelerated exsolution of nanoparticles in non-stoichiometric perovskite thin films

Hyeon Han [1,2,10] ✉, Yaolong Xing[3,10], Bumsu Park [3,4], Dmitry I. Bazhanov[5,6], Yeongrok Jin[7], John T. S. Irvine[8], Jaekwang Lee [7] & Sang Ho Oh [3,9] ✉

Exsolution of excess transition metal cations from a non-stoichiometric perovskite oxide has sparked interest as a facile route for the formation of stable nanoparticles on the oxide surface. However, the atomic-scale mechanism of this nanoparticle formation remains largely unknown. The present in situ scanning transmission electron microscopy combined with density functional theory calculation revealed that the anti-phase boundaries (APBs) characterized by the $a/2 < 011>$ type lattice displacement accommodate the excess B-site cation (Ni) through the edge-sharing of $BO_6$ octahedra in a non-stoichiometric $ABO_3$ perovskite oxide ($La_{0.2}Sr_{0.7}Ni_{0.1}Ti_{0.9}O_{3-\delta}$) and provide the fast diffusion pathways for nanoparticle formation by exsolution. Moreover, the APBs further promote the outward diffusion of the excess Ni toward the surface as the segregation energy of Ni is lower at the APB/surface intersection. The formation of nanoparticles occurs through the two-step crystallization mechanism, i.e., the nucleation of an amorphous phase followed by crystallization, and via reactive wetting on the oxide support, which facilitates the formation of a stable triple junction and coherent interface, leading to the distinct socketing of nanoparticles to the oxide support. The atomic-scale mechanism unveiled in this study can provide insights into the design of highly stable nanostructures.

Exsolution of the excess cations from a non-stoichiometric perovskite oxide has received growing attention as a facile route to form uniformly dispersed nanoparticles (NPs) on the oxide surface for applications in catalysts and renewable energy[1–23]. This method, compared with the conventional deposition methods of NPs[1–3], renders more uniform dispersion of NPs with a narrower size distribution. Moreover,

the socketed NP structure leads to exceptionally high thermal stability and coking resistance owing to the strong interaction between the anchored particle and the oxide support[4]. Additionally, the exsolution process can lead to creation of magnetic nanostructure[12,19] and also significantly improve the conductivity of the host oxide[14,15]. As such, exsolution has emerged as a new platform for the design of supported

[1]Max Planck Institute of Microstructure Physics, Weinberg 2, 06120 Halle (Saale), Germany. [2]Department of Materials Science and Engineering, Pohang University of Science and Technology (POSTECH), Pohang 37673, Republic of Korea. [3]Department of Energy Science, Sungkyunkwan University, Suwon 16419, Republic of Korea. [4]CEMES-CNRS, 29 rue J. Marvig, 31055 Toulouse, France. [5]Faculty of Physics & Faculty of Computational Mathematics and Cybernetics, Moscow State University, GSP-1, Leninskye Gory,1-2, 119991 Moscow, Russia. [6]Federal Research Center "Computer Science and Control" of the Russian Academy of Sciences, FRC CSC RAS, Vavilova St. 44/2, 119333 Moscow, Russia. [7]Department of Physics, Pusan National University, Busan 46241, Republic of Korea. [8]School of Chemistry, University of St Andrews, St Andrews, Fife KY16 9ST, UK. [9]Department of Energy Engineering, KENTECH Institute for Energy Materials and Devices, Korea Institute of Energy Technology (KENTECH), Naju 58330, Republic of Korea. [10]These authors contributed equally: Hyeon Han, Yaolong Xing. ✉e-mail: hyeonhan21@gmail.com; shoh@kentech.ac.kr

heterogeneous NPs on functional oxides. Various efforts have been made to improve the exsolution process, including the electrochemical poling[1], metal-oxygen bond strength control[6], facet-dependent interfacial energy control[7], co-segregation from double perovskite lattices[8], and charge-balanced doping[15].

The thermodynamic driving force for the exsolution is provided by lowering the chemical potential of cation and oxygen in off-stoichiometric perovskite oxide. Namely, for an $A_{1-a}BO_{3-x}$ perovskite oxide where both A-site cation and oxygen are deficient, a high temperature annealing at reducing atmosphere can effectively trigger the exsolution of the B-site cations to restore the $ABO_3$ stoichiometry[3]. While the thermodynamic driving force for the exsolution is clear, the mechanism that leads to the preferential formation of NPs on top of the oxide surface is not clearly understood. Some in situ transmission electron microscopy studies have been performed to explore the NP formation process in the surface area using bulk polycrystalline samples[22,23]. However, to elucidate the preferential nucleation site and the diffusion pathway through which excess B-site cations reach the surface, in situ cross-sectional imaging of single crystalline samples with well-defined facets are highly needed. Moreover, considering a large energy barrier for the bulk diffusion, which is in the range of 3-4 eV[24], there must be energetically favored diffusion pathways to the surface other than the bulk diffusion.

Off-stoichiometry often drives the formation of extended structural defects in perovskite oxides. The defects associated with the accommodation of cation and oxygen deficiency, such as stacking faults or anti-phase boundaries (APBs), may present low energy segregation site and/or diffusion path for the excess cations to reach the surface. The well-known Ruddlesden-Popper (RP) planar faults are consistently observed in A-site rich perovskites[24,25]. The fast diffusion of the A-site cation along the RP planar faults leading to the resistive switching has been demonstrated for the A-site rich perovskite[24]. In the case of A-site deficient perovskites ($A_{1-x}BO_3$), defects can be considered as related to antiphase defects. In the archetype tetragonal tungsten bronze ($A_{0.6}BO_3$) structure, Thomas demonstrated how the defects can be considered as a rotation antiphase boundary[26]. In thin films of A-site deficient perovskites, also termed B-site rich perovskites, anti-phase domains (APDs) can be formed, which are bounded by the APBs on the {100} and {010} planes[27]. The APBs are related by the lattice displacement vector of $a/2<011>$ with respect to surrounding matrix, consisting of edge-shared $BO_6$ octahedral arrangement. It has been shown that the APBs can also provide the fast diffusion path for off-stoichiometric cations[24]. Thus, the extended structural defects can play a crucial role in promoting particle nucleation/migration and thereby guide the microstructure optimization.

Another important issue is how the nucleation and crystal growth of NPs occurs from the diffused species, resulting in such strong interaction with the oxide support. This is a crucial feature that leads to the exceptionally high thermal stability of exsolved NPs[4]. In general, it has been found that polymorphism, the ability of a system to exist in different crystalline structures, is an important phenomenon in the formation of NPs[28–31]. According to Ostwald's rule of stages, the first occurring phase in polymorphism is normally the one which is closest in free energy to the mother phase, that is, the least stable phase, followed by phases in order of increasing stability. An intriguing example of Ostwald's rule is the so-called two-step crystallization (TSC) of NPs. According to the TSC model, instead of the direct nucleation of crystalline phase, a stable nucleus first evolves as an amorphous phase and the crystallization follows within the amorphous nucleus. Considering that NPs forming on oxide film surface are subject to the force balance as well as lattice matching at the triple junction with oxide, the formation of NPs via TSC is certainly beneficial as the constituent atoms can easily migrate and adjust themselves in amorphous state to find stable position during the crystallization. However, the TSC

mechanism has not been exploited for the exsolution process of NP formation.

Here, we present direct observation of the exsolution process of NP formation from epitaxial non-stoichiometric perovskite oxide films using in situ scanning transmission electron microscopy (STEM), revealing the most efficient microstructural feature governing the exsolution of NPs to the film surface. The epitaxial thin film of $La_{0.2}Sr_{0.7}Ni_{0.1}Ti_{0.9}O_{3-\delta}$ (LSNT) with 10% A-site cation and controlled oxygen deficiency was grown on $(LaAlO_3)_{0.3}(Sr_2AlTaO_6)_{0.7}$ (LSAT) (001) substrate by using pulsed laser deposition (PLD). We found that the non-stoichiometric epitaxial film contained APBs running across the film thickness along the {110} planes, which accommodate the oxygen deficiency by changing the corner-shared oxygen octahedra arrangement locally to the edge-shared arrangement. The in situ cross-sectional STEM observation resolving the exsolution process across the film thickness to the surface, in conjunction with density functional theory (DFT) calculations, reveal that the APBs are the key structural feature promoting the exsolution of Ni NPs as they provide energetically favored segregation sites for Ni ions due to the low segregation energy. Furthermore, we observed that the APBs provide fast diffusion pathways towards the surface, facilitating the exsolution of NPs predominantly at their intersection with the film surface. Finally, our in situ atomic-resolution imaging captured that the formation of NPs occurs through the TSC mechanism which facilitates the formation of stable triple junction and coherent interface with oxide support, leading to the distinct lattice-matched socketing of NPs to the oxide support.

## Effects of non-stoichiometry on the exsolution

To verify the nature of defects induced by the non-stoichiometry and their roles in the exsolution, we prepared the controlled non-stoichiometric as well as stoichiometric LSNT films for comparison. The strain state and microstructure of the two films with a thickness of 100 nm were characterized by various techniques including X-ray diffraction, electron diffraction (SAED), atomic-scale STEM high-angle annular dark field (HAADF) imaging, and subsequent geometric phase analysis (GPA) (Figs. 1a, b and Supplementary Fig. 1). The results show that both films are fully strained without relaxation by misfit dislocations. After the annealing at 900 °C in reducing atmosphere (-1 × $10^{-6}$ Torr), the films remained fully strained but the exsolution of Ni NPs from the perovskite lattice resulted in the contraction of the out-of-plane lattice parameter. The extent of the lattice contraction was larger in the non-stoichiometric film than the stoichiometric film, implying more Ni atoms are exsolved from the perovskite lattice. The STEM-HAADF images and their GPA strain maps showed no misfit or threading dislocation formation in the annealed films, excluding the role of dislocation in the exsolution as a preferential path for diffusion.

The Ni NPs formed via the exsolution process of the two systems were characterized by scanning electron microscopy (SEM) surface imaging of the annealed films (Fig. 1c and d). The quantitative measurement and statistical analysis of the Ni NPs images yields a much higher areal density (795 $\mu m^{-2}$) with smaller size (8.8 nm) for the non-stoichiometric film. For the stoichiometric film, the corresponding values are 51 $\mu m^{-2}$ and 10.7 nm, respectively. The different extent of the exsolution of Ni was further confirmed by X-ray photoelectron spectroscopy (XPS) by the deconvolution of the mixed valence states of Ni detected from near-surface region (Supplementary Fig. 2). The $Ni^{3+}$ and $Ni^0$ states are attributed to the Ni atoms bonded to the perovskite lattice and the exsolved Ni particle, respectively. The fraction of $Ni^0$ in total Ni is determined to be 66% and 10% for the non-stoichiometric and the stoichiometric films, respectively. The SEM and XPS results reveal the crucial role of non-stoichiometry in the formation of Ni NPs by exsolution. Cross-sectional STEM-HAADF imaging and energy-dispersive X-ray spectroscopy (EDS) elemental mapping have been carried out to visualize the elemental distribution of Ni and correlated

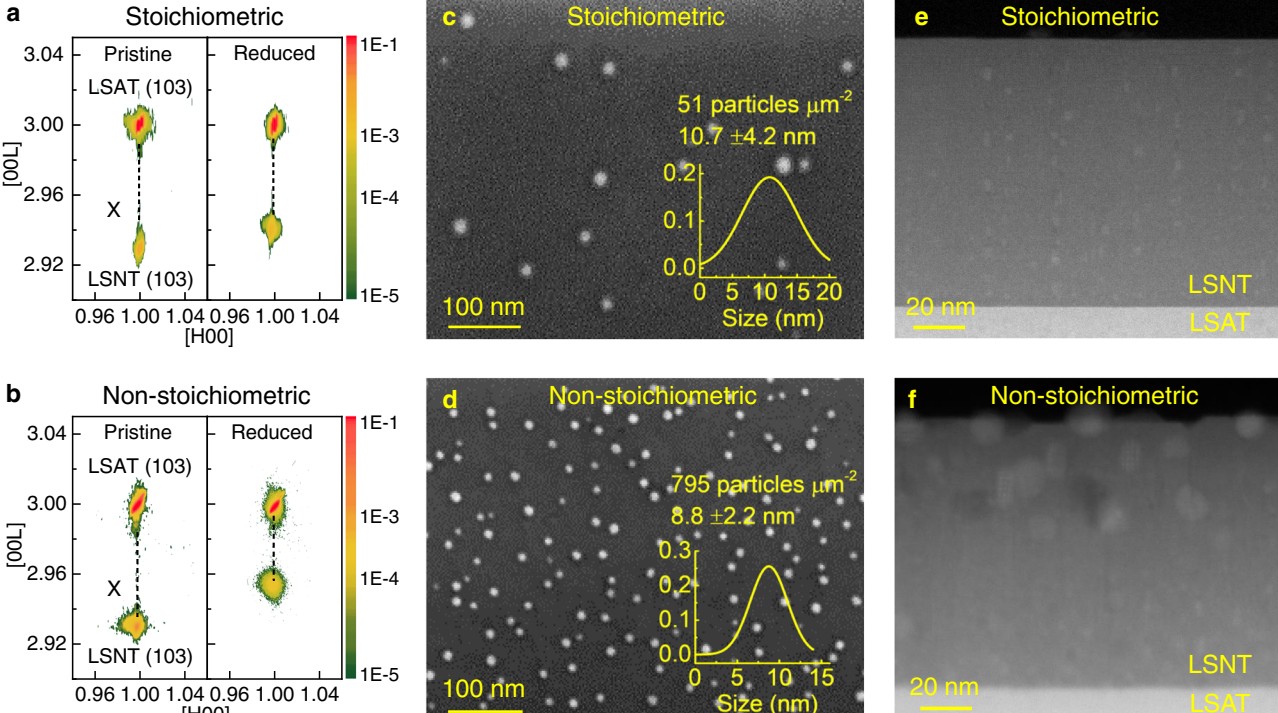

**Fig. 1 | Effect of non-stoichiometry of perovskite oxide on NP formation by exsolution.** Reciprocal space maps before and after reduction for **a** the stoichiometric and **b** non-stoichiometric LSNT film. The film reduction was performed in a vacuum furnace (-1 × 10⁻⁶ Torr) at 900 °C for 10 h. All films show that the reciprocal lattice points of film and substrate are aligned along the same [H00], indicating a fully strained state after the reduction. The reciprocal lattice points of the bulk LSNT (103) are marked with "X". SEM surface images after reduction of **c** the stoichiometric and **d** the non-stoichiometric LSNT film. Cross-sectional STEM-HAADF images after reduction of **e** the stoichiometric and **f** the non-stoichiometric thin film. The thickness of the films is 100 nm.

microstructural features in the annealed films (Fig. 1e, f and Supplementary Fig. 3). Compared to the stoichiometric film, a higher density of Ni NPs was detected even inside the non-stoichiometric film. Moreover, Ni NPs tend to align vertically along the extended defects formed in the non-stoichiometric film.

**Anti-phase boundaries in non-stoichiometric film**

The non-stoichiometric LSNT film exhibits a high crystalline quality except for the defects running vertically from the interface to the surface. The observed extended defects are APDs formed by a lattice translation vector of a/2 < 011> with respect to the surrounding matrix; the atom-resolved EDS La map clearly shows that the La columns from two adjacent domains are related by a projected lattice translation vector of a/2[001] (Fig. 2a–c and Supplementary Fig. 4). The atomic structure of the resulting APB is found to consist of the chain of edge-sharing $TiO_6$ octahedra, implying the local enrichment of B-site cations per formula unit (Fig. 2d–f). In fact, the EDS Ni map reveals the strong enrichment of the APBs by Ni (Fig. 2c) (note that for the APD embedded within the matrix the pronouncing Ni signal is along the two vertical (100) edge-on APBs marked by white dash lines).

To identify the origin of the Ni enrichment of APB, we calculated the segregation energy ($E_{seg1}$) of Ni, which is defined as the difference in the total free energy between a APB site and a bulk-like site for each ion substitution. This means that a positive $E_{seg1}$ suggests a tendency to move into the bulk, while a negative $E_{seg1}$ implies Ni segregation at APB. The $E_{seg1}$ of Ni at the APB site in the edge-sharing octahedra is calculated to be −0.17 eV (Fig. 2e), indicating that the substitution of Ti ions in the APBs with Ni is energetically favored. Considering that the high temperature film growth process, the Ni enrichment of APB is energetically plausible due to the thermal activation.

The solute segregation at defects such as dislocations, grain boundaries, free surface and APBs is a phenomenon that is ubiquitous in various ionic solids as well as metals[32–35], attributed to various factors, including surface free energy, elastic strain energy, and electrostatic potential[33–35]. In the case of ionic solids, the segregation is influenced strongly by the electrostatic potential difference between the host and the solute ions[34,35]. In the present case, the valence state of the solute ($Ni^{2+}$) ion is largely different from the host B-site cation ($Ti^{4+}$). However, given that the APB with edge-sharing octahedra accommodates the oxygen deficiency and, as a consequence, the valence state of B-site cation is reduced, the segregation of a cation with reduced valence state can be favored at APBs. The negative $E_{seg1}$ of Ni to the Ti at the APB may have this electrostatic origin.

The existence of edge-sharing feature for $TiO_6$ octahedra can be considered to provide the flexibility for changing their directions, thus for forming enclosed antiphase domains. The APBs lie mostly in the {100} planes and partially in the {110} planes. In the atomic-resolution STEM-HAADF images, due to overlap of the antiphase nanodomains with the surrounding matrix along the viewing direction, the domains usually appear as a complicated lattice pattern with additional atomic columns with different intensity levels. The simulated STEM image using a structural model consisted of two domains related by antiphase lattice translation by a/2 < 011> perfectly reproduces the key structural features of the experimental image. The cross-correlation factor (XCF) of -0.96 was obtained for the simulated image (XCF of 1.0 indicates perfect match) as shown in Supplementary Fig. 5.

The change of the octahedra arrangement from corner- to edge-shared at the APB is effective in accommodating not only excess B-site cation but also oxygen deficiency. This local change in the oxygen coordination results in the change of the electronic structure, which can be assessed by electron energy-loss spectroscopy (EELS) (Fig. 2g–i). The EELS signals from the APD include the overlapping APBs and matrix, which can be compared with those from the matrix free of APB. For comparison of the two EELS data, the energy

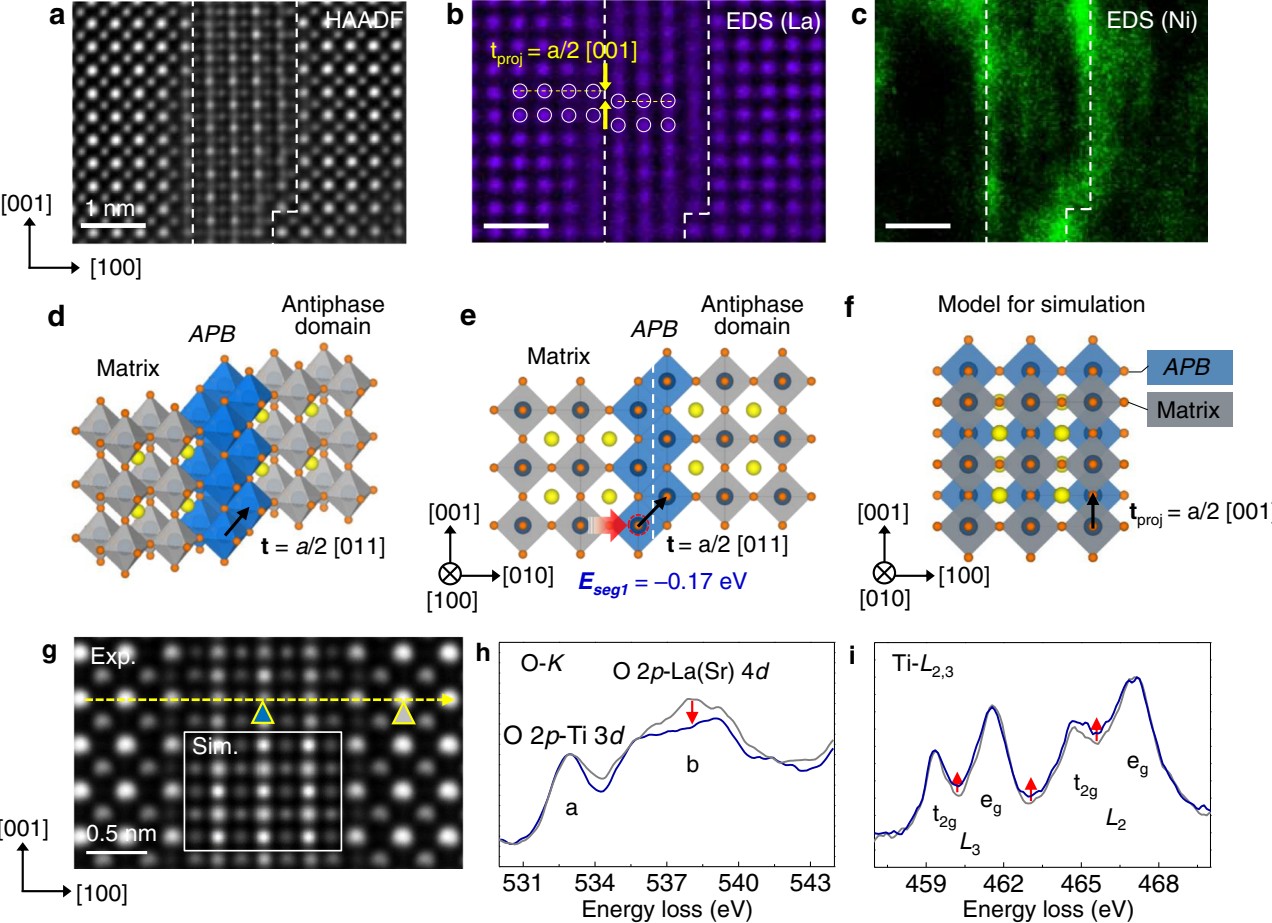

**Fig. 2 | Ni enrichment of APBs in the pristine non-stoichiometric LSNT film. a** A STEM-HAADF image and the corresponding **b** La and **c** Ni EDS elemental map of a typical APB, indicating the Ni enrichment of the APB in the pristine non-stoichiometric 60 nm thick film. The atomic-resolution La EDS map confirms the proposed APD structure with the projected translation vector of $a/2$ [001]. **d–f** Atomic structure model of APB with the lattice translation vector of $a/2$ [011]. The yellow, navy, and orange spheres denote A-site (La/Sr), B-site (Ni/Ti) cations, and O ions, respectively. The gray octahedra represent the core-shared network in the matrix and APD; the blue octahedra represent the edge-shared network in the APB. The white dashed line denotes APB. The calculated segregation energy ($E_{seg1}$) of Ni to the B-site at the APB is shown in **e** which is −0.17 eV. The projection of the model along the [010] zone axis in **f** where the matrix, APD and APB are overlapped, was used for STEM image simulation. **g** Simulated STEM-HAADF image using the APB structural model. EEL spectra of **h** O-$K$ edge and **i** Ti-$L_{2,3}$ edge. The measurement locations for the matrix perovskite and the APBs are indicated by gray and blue triangular marks, respectively, in **g**. The blue and gray spectra represent the spectra obtained from the perovskite matrix and the APB, respectively. The red arrows highlight the difference between the matrix and the APB in fine structure of EEL spectra due to the oxygen deficiency and the related change of Ti valence state. Source data are provided as a Source Data file.

was calibrated by using zero-loss peak and the intensity was normalized with respect to the strongest peak (the whole spectra is shown in Supplementary Fig. 6). The EELS O-$K$ edge (Fig. 2h) obtained across the antiphase domain shows that the post-peak (marked by 'b') in the domain is suppressed compared to the pre-peak (marked by 'a'). This change in the EELS O-$K$ edge originates from the reduced oxygen coordination due to the edge-shared octahedra at APB.

The EELS Ti-$L_{2,3}$ edge (Fig. 2i) consists of two major peaks $L_3$ and $L_2$ attributed to the spin-orbit coupling in 3$d$ orbitals. The crystal field splitting corresponding to the octahedral coordination results in the further splitting into $t_{2g}$ and $e_g$ sub-bands. At the matrix region, the valence state of Ti is close to 4+, results in well separated $t_{2g}$ and $e_g$ peaks in both $L_3$ and $L_2$ edges[36]. Accompanied by oxygen deficiency, the valence state of Ti ions in the APB is reduced from 4+ to 3+ due to the electron doping from the edge-shared oxygen. In summary, our STEM investigation reveals that the APBs in non-stoichiometric LSNT film accommodate not only the excess Ni but also oxygen deficiency by changing the corner-shared oxygen octahedra arrangement locally to the edge-shared arrangement, thereby acting as a reservoir of excess Ni.

## Anti-phase boundaries as fast diffusion paths

The formation of NPs driven by the exsolution of Ni has been observed in situ by heating a cross-sectional TEM sample at 700 °C in vacuum (~$10^{-7}$ Torr). The STEM images and correlated EDS maps (Fig. 3a–e, Supplementary Figs. 7 and 8) confirm the preferential formation of a high density of NPs along the APDs. As to the diffusion species that leads to the formation of Ni NPs, it is most likely the metal cations (Ni ions). The Ni NPs form not only on the film surface but also inside the film, which is consistent with the ex situ observation of the annealed film (Fig. 1). Correlation of Ni NPs with APDs is illustrated schematically in Fig. 3f. The observed exsolution behavior indicates that the nucleation of new particles is energetically favored over the ripening or coalescence of nucleated particles in the presence of APBs. We propose that the role of APBs in the exsolution process is twofold; it reduces the energy barriers for the migration of Ni and for the heterogeneous nucleation.

To investigate how the APB affects the migration of Ni through the PV lattice, we calculated the energy barrier along the three different pathways by performing nudged-elastic band (NEB) method (Fig. 4a, b). As a reference, the migration of Ni toward the nearest neighboring

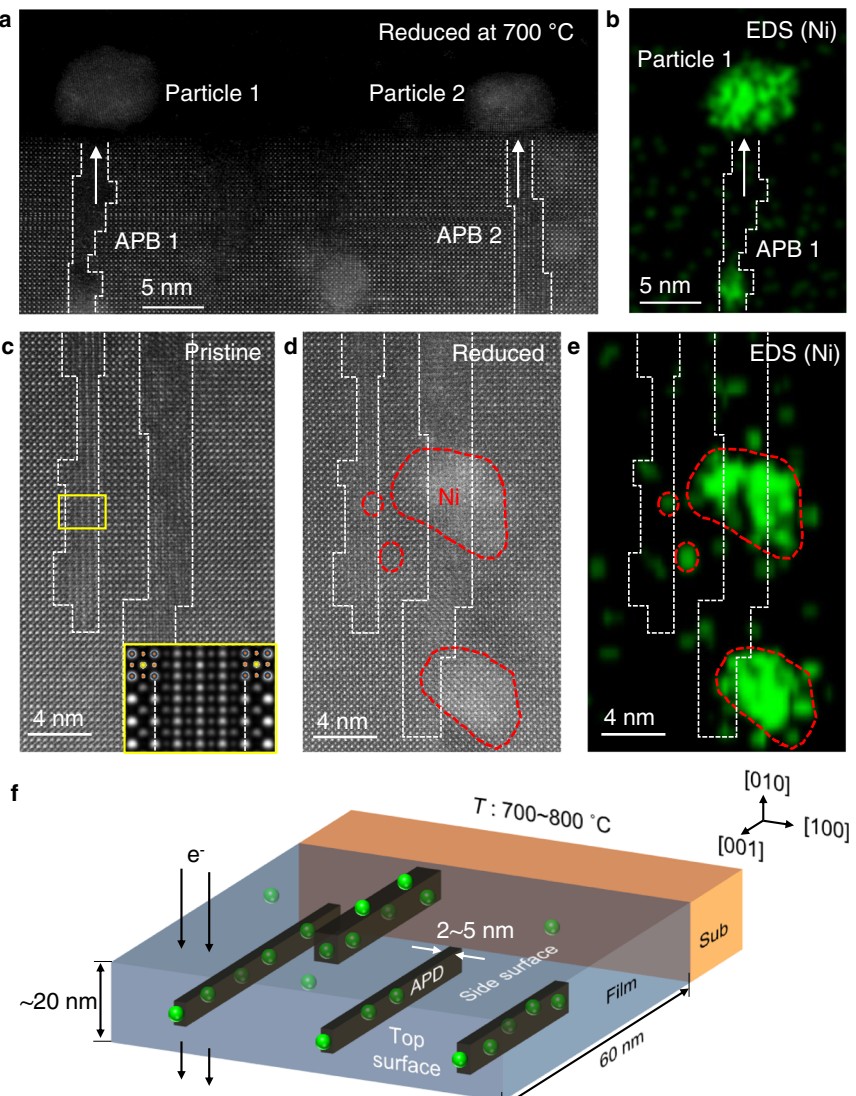

**Fig. 3 | In situ STEM observation of NP formation by exsolution of Ni in the non-stoichiometric LSNT film. a** Cross-sectional STEM-HAADF image and **b** the corresponding Ni EDS map of non-stoichiometric 60 nm thick film after heating inside STEM. Magnified STEM-HAADF images **c** before and **d** after in situ heating. **e** Ni EDS map of **d** showing the preferential formation of Ni NPs at APBs. APBs are outlined by white dashed lines. The inset in **c** represents the simulated APB model in good agreement with the experimental image. The complicated STEM-HAADF image contrast originates from the overlap of APD and matrix along the viewing direction. The in situ heating was performed at 700 °C for 45 min in the high vacuum (~10⁻⁷ Torr) of TEM column. **f** Schematic illustrating the preferential formation of Ni NPs near the APBs as observed by in situ STEM heating. Considering the size of APDs (2–5 nm in width), most of them are embedded within the TEM specimen (~20 nm in thickness).

B-site far from an APB was considered, which yields the energy barrier of 4.37 eV. The energy barrier for a similar path adjacent to the APB is reduced to 3.35 eV (Boundary diffusion 1 in Fig. 4a). On the other hand, the energy barrier for the migration toward the nearest B-site across the APB is calculated to be 0.68 eV (Boundary diffusion 2 in Fig. 4a). This dramatic reduction of the energy barrier for the migration of Ni originates primarily from the different electrostatic interactions of Ni with the surrounding atoms depending on the migration pathways. For the boundary diffusion 2, the migration of Ni is mainly influenced by only neighboring oxygen ions whereas Ni heavily interacts with the surrounding cations as well as oxygen ions simultaneously for bulk and boundary diffusion 1, raising the migration barrier in association with a large lattice distortion. A similar effect of APB in the migration barrier has also been reported for SrTiO₃[24].

Next, we calculated the $E_{seg2}$ (Fig. 4c) of Ni along the APB, defined as the difference in the total free energy between the APB site at the film surface and the internal APB site for each Ni ion substitution (See

Methods for calculation details). The calculated $E_{seg2}$ shows a negative value of −0.54 eV, indicating the existence of the driving force for the outward migration of Ni towards the surface along the APB pathway.

**Two-step crystallization and reactive wetting of nanoparticles**
In situ atomic-resolution STEM imaging revealed that the crystalline state of Ni NP form via TSC rather than the direct nucleation of crystalline Ni (Fig. 5); first the amorphous state of Ni evolves in form of hemi-spherical droplet, and then the crystallization takes place subsequently within the amorphous droplet (See Supplementary Video 1). At an early stage of NP formation, a small amorphous Ni droplet started forming preferentially on top of APD (Fig. 5a) with establishing the contact angle (θ) of about 90° (Fig. 5b). While the amorphous droplet continues growing up to ~3 nm in the equivalent spherical diameter (ESD) (Fig. 5g), the crystalline phase nucleates within the amorphous droplet preferentially at the interface with underlying oxide surface (Fig. 5c). Before being stabilized in the crystalline phase, the sub-

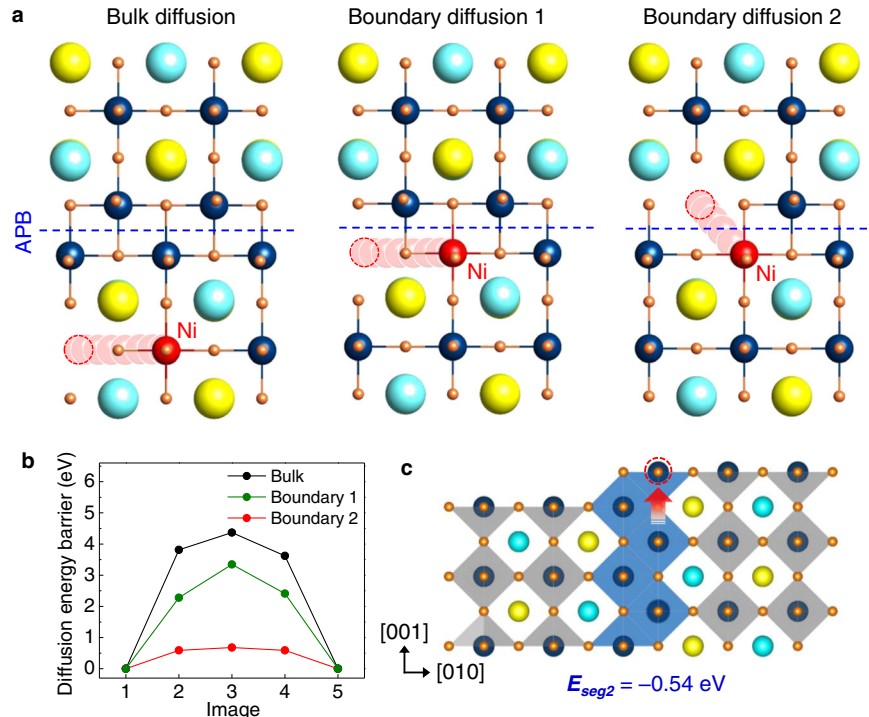

**Fig. 4 | DFT calculation for the migration barriers of Ni in the vicinity of APB.** **a** Three migration pathways; bulk diffusion, diffusion parallel to the APB (Boundary diffusion 1), and diffusion across the APB (Boundary diffusion 2). **b** Energy barriers calculated for the three migration pathways. The migration of Ni toward the nearest neighboring B-site far from an APB yields 4.37 eV. The energy barrier for a similar

path adjacent to the APB is reduced to 3.35 eV (Boundary diffusion). The energy barrier for the migration toward the nearest B-site across the APB is calculated to be 0.68 eV (Boundary diffusion 2). **c** Segregation energy ($E_{seg2}$) of Ni calculated for the APB site at the film surface from the internal APB site. Source data are provided as a Source Data file.

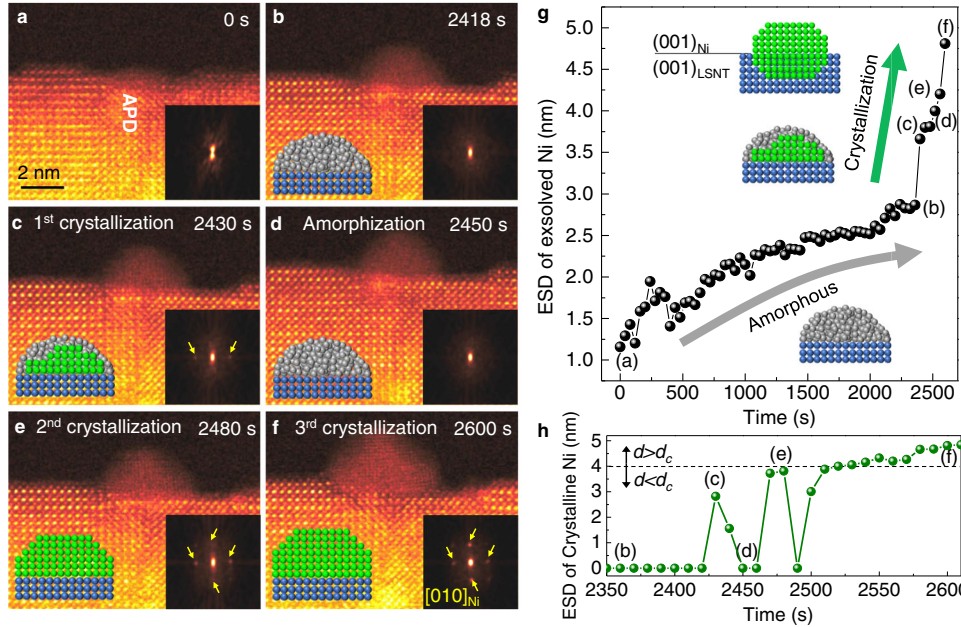

**Fig. 5 | Time-resolved in situ STEM imaging of NP formation process.** **a**–**f** Sequential STEM-HAADF images showing the evolution of crystalline Ni NP on top of APD via TSC. Insets are FFT patterns obtained from the Ni NP. The yellow arrows in FFT patterns indicates the reflections from Ni NP. The in-plane lattice fringes appeared first in **c**. **g** Time-dependent evolution of the size of Ni NP. The equivalent spherical diameter (ESD) of NP was measured directly on time-series of STEM images. The time corresponding to the STEM images in **a**–**f** are denoted accordingly. The amorphous droplet was formed first. Subsequently, the crystalline phase is stabilized after several structural fluctuations. Once the crystalline phase is

stabilized, the ESD of crystalline Ni increases rapidly. The reactive wetting of Ni NP with the oxide results in the formation of ridge at the triple junction. **h** Time-dependent evolution of the size of crystalline Ni (ESD, nm), showing the structural fluctuation via repeated dissolution and crystallization. When the ESD of crystalline Ni ($d$) is smaller than the critical diameter ($d_c$), the crystal dissolves back into the amorphous droplet. If $d$ is larger than $d_c$, the crystalline phase grows rapidly. The obtained $d_c$ is ~3.83 ± 0.08 nm. The in situ reduction was performed at 800 °C in the high vacuum (~10⁻⁷ Torr) on the non-stoichiometric 60 nm thick film. Source data are provided as a Source Data file.

critical nuclei within the amorphous droplets exhibit structural fluctuations, resulting in dissolution and crystallization back and forth several times as confirmed by fast Fourier transformation (FFT) patterns (Fig. 5c–f). Once the crystalline phase reaches the critical size ($d_c$), it becomes stable and grows continuously without structural fluctuations. The temporal evolution of the NP size and the size of the internal crystallite are plotted in Fig. 5g and h, respectively. The $d_c$ of the crystalline phase is measured to be ~3.83 ± 0.08 nm. The TSC of Ni NPs has been observed reproducibly throughout the in situ experiments as shown in Supplementary Fig. 9.

The amorphous state of Ni is energetically favored as the first evolving phase in the NP formation process via TSC because its lower interfacial free energy compared to that of crystalline states can effectively reduce the energy barrier. The structural fluctuations of sub-critical nuclei inside the droplet, i.e., repeated dissolution and crystallization, indicates that the energy barrier for the crystallization of amorphous phase also exists but at a reduced magnitude. Nevertheless, the subsequent nucleation of the stable crystalline phase within the amorphous droplet indicates that the amorphous phase is metastable with respect to the crystalline phase. During the crystallization the in-plane lattice fringes are resolved first (see FFT in Fig. 5c); the in-plane lattice fringes appear from the central region of amorphous droplet at the interface with support and extends towards the edges. The formation of lattice-matched coherent interface can reduce the NP-support interface energy at the expense of the elastic strain energy of NP. By carrying out control experiments at 20 °C (Supplementary Fig. 10), we can exclude the effects of the electron beam on the structural fluctuations occurring during the early stage of NP formation process.

Considering that NPs forming on top of the oxide support are subject to not only the elastic strain energy due to the lattice mismatch but also the surface/interface tension forces at the triple junction, the formation of NPs via TSC is certainly beneficial as the constituent atoms can easily migrate and adjust themselves in amorphous state to find stable position during the crystallization. Furthermore, the amorphous state of NP can easily undergo the reactive wetting with the oxide support during the crystallization, resulting in the formation of a ridge that balances the surface/interface tension forces at the triple junction, that is so-called the socketing of NP into the support (Supplementary Video 2 and Fig. 11). The formation of the ridge along the triple junction occurs through dissolution and redeposition of the oxide material, which is a well-known signature of the reactive wetting of a metallic droplet on oxide substrate, requiring the mass transport through the triple junction[37]. As indicated by yellow arrows in Supplementary Figs. 11c and 11d, a rim of epitaxial oxide layer with ~1-2 unit cell thickness that surrounds the NP was formed. The ridge can balance the surface/interface tension forces not only along the in-plane direction but also along the out-of-plane direction, which enhances the thermal stability of exsolved NPs. After prolonged annealing, the crystalline NP evolves with well-defined facets and the lower half about 50% of volume is submerged with strong anchoring to the perovskite support. After distinct eight-fold facets are developed both outside and inside of the perovskite support, the outside part on top of the support grows fast along the [001] and the [101] direction, while the sideways growth along the [100] direction is suppressed as depicted by white dashed line in Supplementary Fig. 11d. The lattice parameter (3.53 Å) of the exsolved particle is in good agreement with that of the Ni crystal (Supplementary Fig. 11e).

## Summary
The present in situ STEM combined with DFT calculation revealed that the APBs characterized by the $a/2<011>$ type lattice displacement accommodate the excess B-site cation (Ni) via the edge sharing of $BO_6$ octahedra in a non-stoichiometric perovskite oxide and provide the fast diffusion pathways for NP formation by exsolution. The atomic-

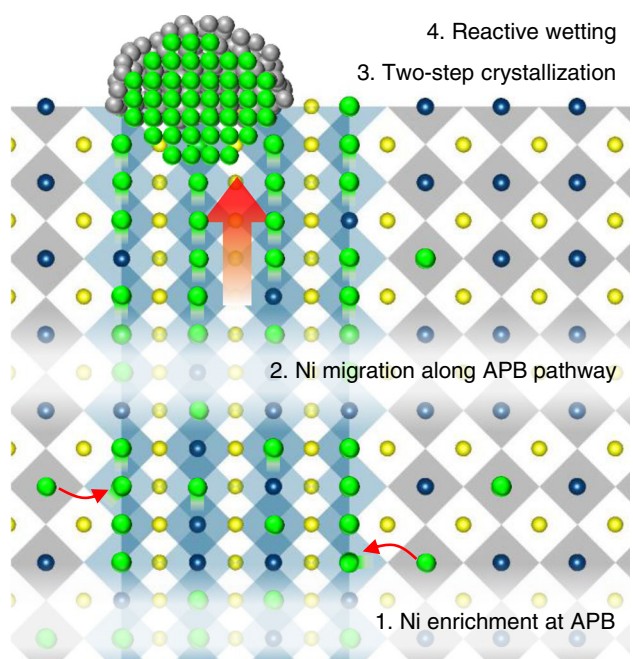

**Fig. 6 | Model proposed for NP formation in a non-stoichiometric perovskite oxide based on in situ STEM observations and DFT calculations.** The APB presents a low energy segregation site and also fast diffusion pathway for exsolved Ni toward surface. The formation of Ni NP occurs via two step crystallization. The reactive wetting of Ni NP with oxide results in the formation of ridge at the triple junction, leading to the strong anchoring of Ni to the oxide. The yellow, green, and navy spheres denote La/Sr (A-site), Ni (B-site), and Ti (B-site) ions, respectively. The gray and green spheres on the surface represent the amorphous and crystalline Ni atoms, respectively. The gray octahedra represent the core-shared network in the matrix and APD; the blue octahedra represent the edge-shared network in the APB.

scale mechanism of the NP exsolution in the non-stoichiometric perovskite oxide is depicted in Fig. 6. The defects are created in the form of APBs which reveal oxygen deficiency and B-cation enrichment due to the edge-sharing of octahedra, when the non-stoichiometric target is used for the film growth. The migration of the excess Ni toward the nearest B-site across the oxygen deficient APB is energetically favored as the anti-site ion configuration greatly reduces the migration barrier. The APBs further promote the outward diffusion of the excess Ni toward the surface as the segregation energy of Ni to the APB is much lower at the APB/surface intersection. As such, B-site enrichment, together with oxygen vacancy, plays an important role in the formation of APB and therefore the accelerated exsolution along the APB. The formation of NPs occurs through the TSC mechanism via reactive wetting on oxide support, which facilitates the formation of stable triple junction and coherent interface with oxide support, leading to the distinct socketing of NPs to the oxide support. The lattice matching and socketing of NP represents the origin of the unprecedentedly high thermal stability of the exsolved NPs compared to the conventional NPs deposited on an oxide support.

In future work, we expect that controlling the degree of the A/B non-stoichiometry can allow wide tuning of the exsolution behavior in thin film forms. For bulk perovskite systems, rational design of new oxide structures having extended defects such as APBs can also lead to the promoted exsolution via controlled formation of nanostructures. Indeed, the formation of cylindrical APDs were already reported in the mixed oxide system[26]. The atomic-scale mechanism unveiled in this study can advance the fundamental understanding on the NP formation process by exsolution and can be used in catalytic and energy applications such as thin film oxide fuel cells and electrolysis cells. Moreover, the extended defect-enhanced exsolution can be employed

to control electronic and magnetic properties of host oxides and the exsolved metal particles, respectively.

## Methods

### Thin film preparation

The non-stoichiometric thin films were grown on LSAT (001) substrates using a non-stoichiometric $La_{0.2}Sr_{0.7}Ni_{0.1}Ti_{0.9}O_{3-\delta}$ (LSNT) target having 10% A-site deficiency by PLD. A stoichiometric LSNT film was also grown using a stoichiometric $La_{0.3}Sr_{0.7}Ni_{0.1}Ti_{0.9}O_3$ target. The oxygen partial pressure, deposition temperature, laser fluence, and repetition rate were fixed at 50 mTorr, 700 °C, 1.5 J cm$^{-2}$, and 5 Hz, respectively. The 100 nm and 60 nm thick films are used for ex situ and in situ TEM measurements, respectively. The reason for using a thinner film for in situ measurements was to image the entire film at a higher magnification. The reduction of the deposited thin films was then performed in a vacuum furnace (-1 × 10$^{-6}$ Torr) at 900 °C with the heating and cooling rates of 5 °C min$^{-1}$.

### Film characterizations

The reciprocal space mapping (RSM) was performed using a high-resolution X-ray diffractometer (D8 Discover, Bruker) under Cu Kα radiation operated at 40 kV and 40 mA. The extent of surface NPs was characterized by a field-emission scanning-electron microscope (JSM − 7800 F PRIME, JEOL Ltd.). Elemental composition and valence states near the film surface were performed using XPS (AXIS Ultra DLD, Kratos. Inc).

### TEM sample preparation

Cross-sectional TEM specimens were prepared along the <010> zone-axis of the epitaxial LSNT thin films grown on LSAT (001) substrates by using Focused Ion Beam (FIB) milling. The prepared lamella by FIB was attached to a Wildfire heating chip for in situ heating experiments. At the final stage of FIB milling, a low energy Ga$^+$ ion beam at 2 kV was used to reduce the beam damages.

### Scanning transmission electron microscopy

A field-emission Cs-corrected (S)TEM (Grand ARM300F, JEOL) operated at 300 kV was used for in situ heating experiments. The in situ heating were performed using a heating holder by DENSsolutions.

Atomic-scale characterization during in situ heating experiment was recorded in annular dark field (ABF) and high-angle annular dark field (HAADF) imaging modes. Detector angle ranges of 7.5 to 17 and 70 to 175 mrad were set for these imaging modes, respectively. The convergence semi-angle for forming the focused probe was 23 mrad. The statistical noise floor in all STEM images was removed using local 2-D Wiener filtering implemented using a commercial software (HRTEM Filter Pro, HREM Research Ltd.).

Energy-dispersive X-ray spectroscopy (EDS) was acquired under 36 mrad convergence angle with spectrometer of a detectable area of 100 mm$^2$ in STEM imaging mode. EDS maps with a total number of ~2000 frames were acquired with a speed of 0.655 s per frame with 256 by 256 pixels. The specimen drift was corrected during acquisition. Each elemental map is constructed by integrating the signal from La-Lα, Sr-Lα, Ni-Kα, Ti-Kα and O-Kα characteristic X-rays, respectively. The EDS maps were processed by the average filter to minimize random noise, which do not contain artifact from filtering.

Electron energy-loss spectroscopy (EELS) line scans were recorded with energy ranges of 400 to 600 eV (for O-K edge and Ti-$L_{2,3}$ edges) using an EEL spectrometer (Gatan GIF Quantum ER, USA) with an energy resolution of 0.6 eV, for detecting the oxygen vacancy and valence state of Ti atoms across anti-phase boundary. The energy dispersion and dwell time per pixel are 0.1 eV and 0.5 s, respectively. The loss energy of the core-loss EELS data was calibrated by tracking the energy drift of the zero-loss peak, which was simultaneously recorded with the core-loss data. The electron dose rate is about 10$^6$ e$^-$ nm$^{-2}$ s$^{-1}$. There is no noticeable e$^-$ beam damage during acquisition.

STEM image simulations were performed using the multislice method in the QSTEM software package[38] using the microscope parameters that closely represent the experimental conditions.

### Geometric phase analysis (GPA)

GPA (HREM Research Inc., Japan) software was used for strain mapping of STEM-HAADF images. For determining strain state of epitaxial thin film in in-plane ($\varepsilon_{xx}$) and out-of-plane ($\varepsilon_{zz}$) direction, reference areas were selected on the substrate area of each sample.

### Segregation energy calculations

Our first-principles study of APBs electronic and structural properties of LSTO perovskite was carried out in the framework of the density functional theory (DFT) as it is implemented in the VASP (Vienna ab initio Simulation Package) code[39,40]. The Kohn-Sham equations[41] are solved within a supercell approach considering the periodic boundary conditions and a plane-wave basis set. Expansion of the plane-wave basis set was limited by a cut-off energy of 400 eV to describe the electronic states in the system. The electron exchange and correlation effects have been taken into account using the generalized gradient approximation (GGA) in the framework of Perdew-Burke-Ernzerhof (PBE) treatment[42].

For total energy and force calculations, we employed an all-electron projector augmented wave (PAW) technique[43]. The Monkhorst-Pack scheme[44] was used for k-point sampling of the Brillouin zone (BZ), while the integration over the BZ was performed on Γ-centered 4 × 4 × 1 k-point mesh, using the tetrahedron method with Blöchl corrections[45]. This MP k-point mesh was found to be optimal for all APB model calculations. Gaussian smearing was used with a width of 0.05 eV to determine partial occupancies of electronic states.

The structural relaxations of APBs geometries were performed via a quasi-Newton algorithm, using the exact Hellmann-Feynman forces acting on each atom. The total energies of APB were converged up to 1 meV/atom, while the residual force acting on each atom was less than 0.01 eV/Å. We found that the chosen parameters provide enough reliability in the accuracy of the present calculations. Besides, in order to account for strong on-site Coulomb repulsion among the 3d electrons of LSNT perovskite the rotationally invariant implementation of the GGA + U method was employed, where the effective Coulomb repulsion $U_{eff}$ = U-J was chosen to be 4.6 eV and 6.0 eV for Ti and Ni atomic species, respectively[8].

Based on our TEM images, the APB structure was constructed using the pseudocubic $La_{0.5}Sr_{0.5}TiO_3$ unit cell with optimized $Pm\overline{3}m$ structure (a = b = c = 7.89 Å). To examine the segregation tendency of Ni dopants, we assume such planar models of APBs as the most relevant facets for B-site metal exsolution in LSNT-based perovskites. A fourteen-layered periodic slab was sufficient to describe segregation phenomenon at APBs.

For this purpose the segregation energies of B-site metal dopant was defined as:

$$E_{seg1} = E_{B-metal}^{Internal\,APB} - E_{B-metal}^{Bulk} \qquad (2)$$

$$E_{seg2} = E_{B-metal}^{Surface\,APB} - E_{B-metal}^{Internal\,APB} \qquad (3)$$

where $E_{B-metal}^{Surface\,APB}$, $E_{B-metal}^{Internal\,APB}$, and $E_{B-metal}^{Bulk}$ are the total energies of the system with B-site metal dopant located on top of APB (at the film surface), the internal APB, and the bulk perovskite, respectively. According to this definition, a negative value of $E_{seg1}$ means easier B-site metal segregation at the internal APB, while a more negative $E_{seg2}$ represents favorable metal exsolution towards the surface of APB.

## Migration energy calculations

A 164-atom supercell was constructed to model APB-containing $La_{0.5}Sr_{0.5}TiO_3$. Initial lattice parameters of the supercell are $a = b = 7.89$ Å and $c = 31.56$ Å with nine $TiO_2$ layers. Two APBs per supercell were considered. Density functional theory (DFT) calculations were performed to calculate the Ni migration barrier in APB-containing $La_{0.5}Sr_{0.5}TiO_3$ with $Ni_{Ti}$ and $V_{Ti}$ using the Vienna Ab Initio Simulation Package (VASP)[40]. The projected augmented wave (PAW) method[43] was used to approximate the electron-ion potential. Exchange-correlation effects were treated within the Perdew-Burke-Ernzerhof (PBE)[42] functional form of the generalized gradient approximation (GGA). A 450 eV cut-off energy for the plane wave basis set and $2 \times 2 \times 1$ Γ-centered $k$-point meshes were used. The calculations were converged in energy to $10^{-6}$ eV/cell and the structures were allowed to relax until the forces were less than $10^{-2}$ eV/Å. Minimum energy diffusion pathways of Ni ions were determined by the climbing image nudged elastic band (CI-NEB) algorithm[46].

## Data availability

The authors declare that the main data supporting the results in this study are available within the article and supplementary files, which is also available from the corresponding authors upon request. Source data are provided with this article. Source data are provided with this paper.

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

## Acknowledgements

This work was supported by National Research Foundation of Korea (NRF) grant funded by the Korea government (MSIT) (No. NRF-2020R1A2C2101735), Creative Materials Discovery Program (NRF-2019M3D1A1078299), the Samsung Research Funding & Incubation Center of Samsung Electronics under Project Number SRFC-MA1702-01, and the KENTECH Research Grant (KRG2022-01-019). D.I.B. acknowledges the financial support from Russian Foundation for Basic Research under Grant No. 19-29-03051MK. The first-principle calculations were performed using the facilities of the Joint Supercomputer Center of the Russian Academy of Sciences (JSCC RAS). J.L. acknowledges the support of an NRF grant funded by the Korean government (NRF-2018R1A2B6004394). J.T.S.I. thanks the EPSRC for support on emergent nanomaterials through Grant EP/R023522/1. Y.X. and S.H.O. acknowledge the support from Advanced Facility Center for Quantum Technology.

## Author contributions

H.H. and S.H.O. conceived the project. H.H. grew and reduced films. H.H. performed XRD, SEM, and XPS measurements. Y.X. performed TEM, EDS, and EELS measurements under the guidance of S.H.O. H.H., Y.X., B.P., J.T.S.I., and S.H.O. discussed the experimental results. D.I.B. performed segregation energy calculations. Y.J. and J.L. carried out migration energy calculations. H.H., Y.X., and S.H.O. wrote the manuscript with input from all other authors.

## Funding

## Competing interests

The authors declare no competing interests.
