## [Peer Review File · Nature Communications]

REVIEWER COMMENTS

Reviewer #1 (Remarks to the Author):

I hereby report on the manuscript by Han et al. on the role of antiphase domains on the Ni exsolution from model perovskite materials. In general, the quality of the work, also through the application of high-end methods, is very high and the conclusions are sound. Having said that, I would recommend the authors to think about incorporating two main points:

1) While I understand, that the use of model systems is necessary to infer the conclusions as stated in the manuscript, I would appreciate if the authors could try to bridge the gap between the model systems and perovskite materials, that are closer to actual use. I assume that for the latter materials, the formation of the antiphase domains and the mechanism of Ni exsolution is different. In other words, how valid are the conclusions for such materials and what implications does this have for, e.g., catalytic use?

2) The authors in a sub-heading speak about the influence of non-stoichiometry on the exsolution mechanism. As far as I understand, the authors discriminate here between a stoichiometric and one non-stoichiometric film. Can the authors comment on how the formation of the antiphase domains and the Ni exsolution actually change as a function of the non-stoichiometry, i.e. steered by A and/or B-site doping?

Reviewer #2 (Remarks to the Author):

The article by Han et al. discusses the preferential exsolution of Ni at anti-phase boundaries or extended defects in sub-stoichiometric perovskite oxide thin films. The noteworthy results include STEM HAADF with EDS analysis of stoichiometric and thin films after reduction at 900 C, showing that Ni exsolution is more pronounced at APBs and also in non-stoichiometric thin films. The implications stated in the article suggest that the exsolution is due to the lower barrier to form the APB extended defects in non-stoichiometric thin films promoting Ni transport and exsolution. Additionally, another noteworthy outcome was the use of in situ TEM imaging which showed the amorphous-crystalline transition of Ni nanoparticles and the "socketing" of the bulk perovskite with the TPB of the Ni nanoparticle.

The work provides additional knowledge on how exsolution of nanoparticles is influenced by the defect concentration of the parent perovskite oxide. It has been known that A sub-stoichiometry promotes nanoparticle exsolution. However this study takes a step further in understanding in why this is the case. The theories developed and conclusions are supported by the experimental experiments and comparisons to related literature. As presented, the work does not have any inherent flaws that would prohibit publication.

Reviewer #3 (Remarks to the Author):

This article provides new insights in the exsolution process of the non-stoichiometric perovskite. In situ microscopy and spectroscopy characterizations clearly reveal the atomic-scale dynamic

structure evolution. The results are very interesting, however, there are some issues that need to be resolved before acceptance for publication.

1. In Figure 1e and 1f, the thickness of the films has distinct difference between stoichiometric and nonstoichiometric films (~ 105 nm and ~ 40 nm). The slow diffusion speed of cations limits the overall extent of exsolution. If the film is too thick, the bulk nucleation may have advantages over diffusion to the surface (Figure 1e). I am wondering if the experimental results are comparable, even through A-site defective perovskites indeed promote the nanoparticles exsolution.

2. APBs in a non-stoichiometric perovskite provide the fast diffusion pathways for nanoparticle formation. Does the oxygen vacancy density play a key role? Is there any difference in oxygen vacancy formation energy between the APB and the bulk?

3. Can the author deduce the diffusion species? Metal cluster, metal cations or oxide?

4. In Figure 5, the structural fluctuations of sub-critical nuclei inside the droplet repeated dissolution and crystallization during the stable crystalline phase formation, which may be a new insight into the exsolution process. I am wondering whether there is an electron beam effect on the structural fluctuations of the unstable nanoparticles? Is there a comparative experiment to confirm this result?

5. There are further a few minor points:

(1): "STEM-HAADF" and "STEM HAADF" should be unified.

(2): In page 9, (marked by 'b') should be (marked by 'a')

RESPONSE TO REVIEWERS' COMMENTS

Reviewer #1

Comment 1: I hereby report on the manuscript by Han et al. on the role of antiphase domains on the Ni exsolution from model perovskite materials. In general, the quality of the work, also through the application of high-end methods, is very high and the conclusions are sound. Having said that, I would recommend the authors to think about incorporating two main points:

While I understand, that the use of model systems is necessary to infer the conclusions as stated in the manuscript, I would appreciate if the authors could try to bridge the gap between the model systems and perovskite materials that are closer to actual use. I assume that for the latter materials, the formation of the antiphase domains and the mechanism of Ni exsolution is different. In other words, how valid are the conclusions for such materials and what implications does this have for, e.g., catalytic use?

• **Reply:** We thank the reviewer for appreciating our work high.

In the manuscript we revealed that the exsolution is accelerated by the APBs present in non-stoichiometric thin films. As the reviewer mentioned, the thin film system we employed in this study may be different from the bulk perovskite systems in actual use. For bulk perovskite systems, we expect designing new oxide structures having extended defects such as APBs can also lead to the promoted exsolution. Indeed, the formation of cylindrical APBs were already reported in the mixed oxide system. (Thomas, J. M. Review Lecture: Topography and Topology in Solid-State Chemistry. *Phil Trans. Royal Soc. London. Series A, Math & and Phys. Sci.* 277, 251-286 (1974).)

Although the bulk perovskite systems can be more representative for applications, using thin film form has advantage in exploring new phenomena which cannot be exploited with the bulk systems, such as enhanced exsolution via substrate-induced strain (*Nat Commun* 2019, 10, 1471.), facet-dependent exsolution (*J Am Chem Soc* 2019, 141, 7509-7517), metal-oxygen bond strength control (*Energ Environ Sci* 2020, 13, 3404-3411), and so on. In practice, the thin film form we employed in this study can be used for thin film oxide fuel cells/electrolysis cells and also in the electronic and magnetic device applications due to the change in conductivity of perovskite and formation of magnetic nanoparticles via exsolution (*Adv. Funct. Mater.* 2022, 32, 2108005; *Nano Lett* 2020, 20, 3538-3544). We have added sentences in the **Summary** part regarding future perspectives regarding rational design of defect-incorporated bulk materials to improve the exsolution behavior the future aspects of our study:

"For bulk perovskite systems, we expect rational design of new oxide structures having extended defects such as APBs can also lead to the promoted exsolution via controlled formation of nanostructures. Indeed, the formation of cylindrical APBs were already reported in the mixed oxide system²⁶."

"The atomic-scale mechanism unveiled in this study can advance the fundamental understanding on the NP formation process by exsolution and can be used in catalytic and energy applications such as thin film oxide fuel cells and electrolysis cells. Moreover, the extended defect-enhanced exsolution can be employed to control electronic and magnetic properties of host oxides and the exsolved metal particles, respectively."

Comment 2: The authors in a sub-heading speak about the influence of non-stoichiometry on the exsolution mechanism. As far as I understand, the authors discriminate here between a stoichiometric and one non-stoichiometric film. Can the authors comment on how the formation of the antiphase domains and the Ni exsolution actually change as a function of the non-stoichiometry, i.e. steered by A and/or B-site doping?

• **Reply:** The use of the non-stoichiometric target ($\text{La}_{0.2}\text{Sr}_{0.7}\text{Ni}_{0.1}\text{Ti}_{0.9}\text{O}_{3-\delta}$ (LSNT) having 10 % A-site deficiency) for the film growth leads to the APB formation in the films, which inherently shows B-site enrichment and oxygen deficiency. The formation of APBs also results in the enrichment of Ni at APBs and the promotion of the subsequent particle migration through the APB pathways. It would be interesting to see the A/B ratio dependent exsolution behavior in the thin films as a function of the non-stoichiometry. We expect that formation of APBs and degree of exsolution will be correlated well with the degree of the non-stoichiometry. However, for these experiments, extensive works are needed by making targets with controlled non-stoichiometry and growing epitaxial thin films using the targets. Thus, we would like to leave this idea for future work. Because we have not observed APBs when the film is grown by using the stoichiometric target, we believe that our results are clear to demonstrate the role of APBs that promote the exsolution behavior in non-stoichiometric films. We have added sentences in the **Summary** part regarding the future work on the control of degree of non-stoichiometry: *"In future work, we expect that controlling the degree of the A/B non-stoichiometry can allow wide tuning of the exsolution behavior in thin film forms."*

Reviewer #2

Comment 1: The article by Han et al. discusses the preferential exsolution of Ni at anti-phase boundaries or extended defects in sub-stoichiometric perovskite oxide thin films. The noteworthy results include STEM HAADF with EDS analysis of stoichiometric and thin films after reduction at 900C, showing that Ni exsolution is more pronounced at APBs and also in non-stoichiometric thin films. The implications stated in the article suggest that the exsolution is due to the lower barrier to form the APB extended defects in non-stoichiometric thin films promoting Ni transport and exsolution. Additionally, another noteworthy outcome was the use of in situ TEM imaging which showed the amorphous-crystalline transition of Ni nanoparticles and the “socketing” of the bulk perovskite with the TPB of the Ni nanoparticle.

The work provides additional knowledge on how exsolution of nanoparticles is influenced by the defect concentration of the parent perovskite oxide. It has been known that A sub-stoichiometry promotes nanoparticle exsolution. However this study takes a step further in understanding in why this is the case. The theories developed and conclusions are supported by the experimental experiments and comparisons to related literature. As presented, the work does not have any inherent flaws that would prohibit publication.

- **Reply:** We thank the reviewer for his/her positive evaluation of our manuscript.

Reviewer #3

Comment 1: This article provides new insights in the exsolution process of the non-stoichiometric perovskite. In situ microscopy and spectroscopy characterizations clearly reveal the atomic-scale dynamic structure evolution. The results are very interesting, however, there are some issues that need to be resolved before acceptance for publication.

In Figure 1e and 1f, the thickness of the films has distinct difference between stoichiometric and nonstoichiometric films (~105 nm and ~40 nm). The slow diffusion speed of cations limits the overall extent of exsolution. If the film is too thick, the bulk nucleation may have advantages over diffusion to the surface (Figure 1e). I am wondering if the experimental results are comparable, even through A-site defective perovskites indeed promote the nanoparticles exsolution.

- **Reply:** We appreciate the reviewer for evaluating our work high, and the valuable comments that help improving our manuscript.

Thanks for pointing out the thicknesses of the two films. We have realized that the scale bar in Figure 1f was wrong. It should be 20 nm instead of 10 nm. We apologize for causing this confusion by mistake. As we previously wrote in Methods, we used the film thickness ranging from 60 nm to 100 nm in this study. Specifically, we used 100 nm for *ex situ* measurements and 60 nm for *in situ* measurements. The reason for using a thinner film for *in situ* measurements was to image the entire film at a higher magnification (as shown in Supplementary Fig. 7). **We have corrected the scale bar in Fig. 1f and clarified the thickness of each measurement in Methods.**

Regarding the thickness-dependent exsolution behavior, we previously reported that the degree of the surface exsolution from non-stoichiometric LSNT films tends to decrease with increasing the film thickness. Nevertheless, it still shows vigorous exsolution behavior even though the film thickness is very large up to 1300 nm (*Nat. Commun.* 2019, 10, 1471). The 1300 nm-thick film still shows a fully strained state before reduction (Supplementary Fig. 5 in the *Nat Comm* paper), and the particle population density ($\sim 200 \mu\text{m cm}^{-2}$) after reduction is larger than that ($\sim 50 \mu\text{m cm}^{-2}$) of the 100 nm stoichiometric sample (Fig. 1c in our manuscript). The reported figure is shown below.

It has reported that the giant exsolution in the non-stoichiometric thin films is mainly due to the lattice strain (*Nat. Commun.* 2019, 10, 1471). However, in our manuscript, the stoichiometric film shows much less active exsolution although it reveals a fully strained state (Fig. 1a). Thus, there should be other reasons for the distinct difference between stoichiometric and non-stoichiometric thin films, which is demonstrated by the role of APBs in our manuscript.

Supplementary Fig. 5 in Nat. Commun. 2019, 10, 1471. Thickness-dependent exsolution (dry H₂, 900 °C, 12 h) in thin films. (a) Reciprocal space mapping (RSM) contours of pristine and reduced LSNT films on STO substrates with different film thicknesses. (b) θ -2 θ XRD patterns of pristine and reduced LSNT (110) films on YSZ (001) substrates with different film thicknesses. Here, the reduced LSNT film has a thickness of 1300 nm. Cross-sectional SEM images of (c) 500-nm-thin and (d) 1300-nm-thin LSNT (110) films on STO substrates. Scale bars, 1000 nm. SEM images of the LSNT films grown on (e) LSAT (001) substrates with different film thicknesses of 100 nm, 500 nm, and 1300 nm (from left to right). Scale bars, 200 nm.

Comment 2: APBs in a non-stoichiometric perovskite provide the fast diffusion pathways for nanoparticle formation. Does the oxygen vacancy density play a key role? Is there any difference in oxygen vacancy formation energy between the APB and the bulk?

• **Reply:** When we used the non-stoichiometric target having B-cation-rich and "oxygen vacancies", the film reveals defects in the form of APBs having "oxygen deficiency" compared to the bulk lattice. As shown in Fig. 2e, the APB structure is inherently associated with oxygen deficiency along with the B-site ion enrichment due to the "edge-sharing" of octahedra (Octahedra share oxygen ions with each other, leading to the oxygen deficiency). Because the enrichment of Ni is revealed at the "oxygen efficient" APB, we calculated segregation energy at APB (Fig. 2e) instead of the oxygen vacancy formation energy. Moreover, APB provides fast diffusion of Ni ions due to the low migration energy of Ni through APB (Fig. 4).

Therefore, oxygen vacancy, together with B-site enrichment, plays an important role in the formation of APB and therefore the accelerated exsolution along the APB.

We have modified the **Summary** part as follows: *"The defects are created in the form of APBs which reveal oxygen deficiency and B-cation enrichment due to the edge-sharing of octahedra, when the non-stoichiometric target is used for the film growth. The migration of the excess Ni toward the nearest B-site across the oxygen deficient APB is energetically favored as the anti-site ion configuration greatly reduces the migration barrier. The APBs further promote the outward diffusion of the excess Ni toward the surface as the segregation energy of Ni to the APB is much lower at the APB/surface intersection. As such, B-site enrichment, together with oxygen vacancy, plays an important role in the formation of APB and therefore the accelerated exsolution along the APB "*

Comment 3: Can the author deduce the diffusion species? Metal cluster, metal cations or oxide?

• **Reply:** It is reported that the exsolved particles from the LSNT thin films are Ni ions (*Nat. Commun.* 2019, 10, 1471; *J. Am. Chem. Soc.* 2019, 141, 7509–7517), which was characterized by *d*-spacing of the particles and EDS maps. We also observed that the cluster of Ni ions from the EDS maps and confirmed the lattice parameter of Ni from the STEM image. We have added **Supplementary Fig. 8** for the EDS maps and **Supplementary Fig. 11e** for the STEM image of the exsolved particle as below. As to the diffusion species, therefore, it is most likely the metal cation (Ni ions) not oxide. We added statements in the revised manuscript, which reads:

"As to the diffusion species that leads to the formation of Ni NPs, it is most likely the metal cations (Ni ions)." in the middle of page 10.

"The lattice parameter (3.53 Å) of the exsolved particle is in good agreement with that of the Ni crystal (Supplementary Fig. S11e)." at the top of page 14.

Supplementary Fig. 8. A cross-sectional STEM-HAADF images and the corresponding EDS maps of the non-stoichiometric thin film after *in situ* reduction. (a) A STEM-HAADF image of the surface area (shown in Figs. 3a and 3b) and the corresponding EDS maps of (b) Ni, (c) O, (d) La, (e) Sr, and (f) Ti. (g) A STEM-HAADF image inside the film (shown in Figs. 3d and 3e) and the corresponding EDS maps of (h) Ni, (i) O, (j) La, (k) Sr, and (l) Ti. Only Ni EDS maps correlate well with the observed particles in the STEM images. The *in situ* heating was performed at 700 °C for 45 min in vacuum ($\sim 10^{-7}$ Torr).

Supplementary Fig. 11. Faceted socketing of exsolved Ni particle. **a** and **b**, time sequence images showing socketed Ni particle with preferred facet epitaxially growing on top of LSNT (001) surface. The faceted surfaces are outlined by yellow dashed lines. **c** and **d**, LSNT perovskite lattice raise up and gradually socket $\{100\}_{\text{Ni}}$ surfaces of Ni particle, indicated by arrows. The size change is indicated by yellow (t_0) and white (t_0+346 s) dashed lines. No remarkable growth is recognized below the surface. **e**, A magnified STEM image of the exsolved particle on the surface, where green spheres denote Ni ions in the inset. The lattice parameter (3.53 Å) of the particle is in good agreement with that of the Ni crystal.

Comment 4: In Figure 5, the structural fluctuations of sub-critical nuclei inside the droplet repeated dissolution and crystallization during the stable crystalline phase formation, which may be a new insight into the exsolution process. I am wondering whether there is an electron beam effect on the structural fluctuations of the unstable nanoparticles? Is there a comparative experiment to confirm this result?

• **Reply:** In this study, we first observed two-step crystallization (TSC) and the structural fluctuations during the early stage of exsolution process. After the stable crystallized Ni NP formed, no structural fluctuations were observed during the growth of NP under the identical electron beam irradiation condition. We note that for such particle exsolution from the stable oxide lattice, we need to apply a very high temperature in a reducing atmosphere. Since we used 800 °C at $\sim 10^{-7}$ Torr for the *in situ* measurements in Fig. 5, we expect that the temperature effect is dominant over the electron beam effects for the crystallization and the structural fluctuations of NPs. Indeed, in control experiments (at 20 °C), we have not observed any particle nucleation or crystallization under the exposure to the electron beam for ~ 50 min, as shown in below Supplementary Fig. 10. Therefore, we can safely exclude the effects of the electron beam on the structural fluctuations. We added this information in the revised manuscript, which read: *"By carrying out control experiments at 20 °C (Supplementary Fig. S10), we can exclude the effects of the electron beam on the structural fluctuations occurring during the early stage of NP formation process."* at the top of page 13.

Supplementary Fig. 10. Time sequence STEM images of the pristine non-stoichiometric thin film measured at 20 °C. The film does not show any nucleation or crystallization when exposed to the electron beam at 20 °C for ~ 50 min. The electron beam current for *in situ* imaging the exsolution process is on the

level between 10^7 - 10^8 $e/\text{\AA}^2\cdot\text{s}$ under the consideration of balancing the image contrast and avoiding beam damage effect. Therefore, the observed structural fluctuations and crystallization in Fig. 5 are expected to be due to the temperature effect rather than the electron beam effect.

Comment 5: There are further a few minor points:

(1): “STEM-HAADF” and “STEM HAADF” should be unified.

(2): In page 9, (marked by ‘b’) should be (marked by ‘a’)

• **Reply:** Thanks for pointing out these. We have corrected them.

REVIEWERS' COMMENTS

Reviewer #1 (Remarks to the Author):

The authors have satisfactorily answered my comments and I now recommend publication.

Reviewer #3 (Remarks to the Author):

The authors have revised the manuscript accordingly, and I recommend its acceptance for publication.

RESPONSE TO REVIEWERS' COMMENTS

Reviewer #1

Comment 1: The authors have satisfactorily answered my comments and I now recommend publication.

- **Reply:** We appreciate your recommendation for publication of our manuscript.

Reviewer #3

Comment 1: The authors have revised the manuscript accordingly, and I recommend its acceptance for publication.

- **Reply:** We thank the reviewer for accepting our manuscript.